# A Longitudinal Seroprevalence Study Evaluating Infection Control and Prevention Strategies at a Large Tertiary Care Center with Low COVID-19 Incidence

**DOI:** 10.3390/ijerph18084201

**Published:** 2021-04-15

**Authors:** Lorenz Schubert, Robert Strassl, Heinz Burgmann, Gabriella Dvorak, Matthias Karer, Michael Kundi, Manuel Kussmann, Heimo Lagler, Felix Lötsch, Christopher Milacek, Markus Obermueller, Zoe Oesterreicher, Christoph Steininger, Karin Stiasny, Florian Thalhammer, Ludwig Traby, Zoltan Vass, Matthias Gerhard Vossen, Lukas Weseslindtner, Stefan Winkler, Selma Tobudic

**Affiliations:** 1Division of Infectious Diseases and Tropical Medicine, Department of Medicine I, Medical University of Vienna, 1090 Vienna, Austria; lorenz.schubert@meduniwien.ac.at (L.S.); heinz.burgmann@meduniwien.ac.at (H.B.); matthias.karer@meduniwien.ac.at (M.K.); manuel.kussmann@meduniwien.ac.at (M.K.); heimo.lagler@meduniwien.ac.at (H.L.); felix.loetsch@meduniwien.ac.at (F.L.); christopher.milacek@meduniwien.ac.at (C.M.); markus.obermueller@meduniwien.ac.at (M.O.); zoe.oesterreicher@meduniwien.ac.at (Z.O.); christoph.steininger@meduniwien.ac.at (C.S.); florian.thalhammer@meduniwien.ac.at (F.T.); ludwig.traby@meduniwien.ac.at (L.T.); zoltan.vass@meduniwien.ac.at (Z.V.); matthias.vossen@meduniwien.ac.at (M.G.V.); stefan.winkler@meduniwien.ac.at (S.W.); 2Division of Clinical Virology, Department of Laboratory Medicine, Medical University of Vienna, 1090 Vienna, Austria; robert.strassl@meduniwien.ac.at; 3Department for Conservative Dentistry and Periodontology, School of Dentistry, Medical University of Vienna, 1090 Vienna, Austria; gabriella.dvorak@meduniwien.ac.at; 4Center for Public Health, Department for Environmental Health, Medical University of Vienna, 1090 Vienna, Austria; michael.kundi@meduniwien.ac.at; 5Center for Virology, Medical University of Vienna, 1090 Vienna, Austria; karin.stiasny@meduniwien.ac.at (K.S.); lukas.weseslindtner@meduniwien.ac.at (L.W.)

**Keywords:** COVID-19, occupational health, infection prevention and control

## Abstract

Personal protective equipment and adherence to disinfection protocols are essential to prevent nosocomial severe acute respiratory syndrome coronavirus (SARS-CoV-2) transmission. Here, we evaluated infection control measures in a prospective longitudinal single-center study at the Vienna General Hospital, the biggest tertiary care center in Austria, with a structurally planned low SARS-CoV-2 exposure. SARS-CoV-2-specific antibodies were assessed by Abbott ARCHITECT chemiluminescent assay (CLIA) in 599 health care workers (HCWs) at the start of the SARS-CoV-2 epidemic in early April and two months later. Neutralization assay confirmed CLIA-positive samples. A structured questionnaire was completed at both visits assessing demographic parameters, family situation, travel history, occupational coronavirus disease 2019 (COVID-19) exposure, and personal protective equipment handling. At the first visit, 6 of 599 participants (1%) tested positive for SARS-CoV-2-specific antibodies. The seroprevalence increased to 1.5% (8/553) at the second visit and did not differ depending on the working environment. Unprotected SARS-CoV-2 exposure (*p* = 0.003), positively tested family members (*p* = 0.04), and travel history (*p* = 0.09) were more frequently reported by positively tested HCWs. Odds for COVID-19 related symptoms were highest for congestion or runny nose (*p* = 0.002) and altered taste or smell (*p* < 0.001). In conclusion, prevention strategies proved feasible in reducing the risk of transmission of SARS-CoV-2 from patients and among HCWs in a low incidence hospital, not exceeding the one described in the general population.

## 1. Introduction

The coronavirus disease 2019 (COVID-19) pandemic, caused by severe acute respiratory syndrome coronavirus (SARS-CoV-2), poses a significant challenge to the health care systems around the world. Health care workers (HCWs) are at increased risk of SARS-CoV-2 infection [1,2]. A 3.4 fold-increased risk for highly-exposed HCWs compared to the general community has been reported [1]. Therefore, infection prevention and control measures are of utmost importance.

Regular testing of HCWs can identify a- or pre-symptomatic individuals and may be part of infection prevention and control strategies among HCWs. However, polymerase chain reaction (PCR) tests are often only performed if individuals are symptomatic, although there is evidence that also asymptomatic or oligo symptomatic cases contribute to virus transmission [3,4]. Hence, the evaluations of SARS-CoV-2 specific antibodies provide a more appropriate tool to describe the dynamics of infection. Previously published reports mainly documented the situation in high-incidence countries in which physicians had to deal with significant workload, complicating the evaluation of implemented strategies [2,5]. In those cohorts, highest risk of SARS-CoV-2 infection was associated with inadequate access to personal protective equipment or reused PPE [1].

In Austria, the first SARS-CoV-2 positively tested individuals were reported at the end of February, with numbers increasing to 9974 cases by the end of March. To suppress community transmission, a variety of infection control measures such as disinfection protocols, social distancing and obligatory personal protective equipment have been implemented by the Austrian government (Appendix A). Herewith, Austria achieved a significant reduction in the daily infection rate, from approximately 1040 new cases at the end of March to about 50 new cases per day at the end of April [6].

In the present study, we report the COVID-19 seroprevalence at the Vienna General Hospital, the largest tertiary care center in Austria and reference center for various patient groups at increased risk for SARS-CoV-2 infection covering the period from April to July 2020 [7,8]. To guarantee the medical care of non-COVID-19 patients, the Vienna Hospital Association implemented a multi-staged plan to allocate COVID-19 patients. The Vienna General Hospital was primarily responsible for the medical care of non-COVID-19 patients, with COVID-19 patients only partially admitted under controlled conditions. Nevertheless, 58 COVID-19 patients were treated from March to July 2020. To minimize the risk of infections for patients and HCWs, specific COVID-19 wards were defined, and strict infection control measures were implemented at the end of March 2020 (Appendix A).

This specific strategy, whose personnel thereby only displayed limited occupational exposure to SARS-CoV-2, enabled us to perform a controlled evaluation of infection control and prevention strategies.

## 2. Materials and Methods

### 2.1. Study Design and Participants

This prospective longitudinal single-center study aimed to evaluate the application of established personal protective equipment (PPE) and prevention strategies, to describe the seroprevalence at our center and to point out risk factors for SARS-CoV-2 infection at the Vienna General Hospital by longitudinal measurement of SARS-CoV-2 antibodies. The baseline visit was conducted at the beginning of April 2020, after the first wave of SARS-CoV-2 spread in Austria, and the follow-up visit was performed two months later at a low incidence rate due to successful infection control measures. The study protocol was approved by the Ethics Committee of the Medical University of Vienna, Austria (ECS 1296/2020), and all study-related procedures were conducted according to the declaration of Helsinki.

### 2.2. Inclusion of Health Care Workers

Nurses, physicians, physiotherapists, as well as administrative staff greater 18 years of age were invited in person or by email to participate in our study. After written informed consent was obtained, the participants were asked to complete a questionnaire covering risk-factors for COVID-19 contact (working environment, reported COVID-19 contact, travel history, household risk), history of SARS-CoV-2 PCR-tests, the occurrence of symptoms and general perception of the epidemic. All participants were invited to a second visit after two months. The serological analysis, as well as the questionnaire, were repeatedly performed at both visits.

### 2.3. Formation of Cohorts and Assessment of Risk Profile

Cohorts were defined depending on their working environment. HCWs working on specific COVID-19 wards were defined as high-risk-cohort. All other HCWs were defined as low-risk-cohort (Appendix A). To assess risk factors for COVID-19, risk profiles were defined. Risk factors comprised of (1) working environment (high-risk if contact to COVID-19 patients, the average risk for HCW without COVID-19 contact), (2) usage of PPE (high-risk: unprotected contact was reported, defined as the absence of filtering facepiece (FFP) 2 mask, goggles or face shield, long-sleeved water-resistant gown and at minimum, one pair of gloves at contact with COVID-19 patient, low-risk: no history of unprotected contact), (3) travel history (high-risk: history of a stay in a high-risk area (high-risk areas are highlighted in the Appendix A), low-risk: no history of a stay in a high-risk area), (4) household contact (high-risk: positive cases in family reported, low-risk: no history of positive cases in family) and (5) problems handling PPE.

### 2.4. Laboratory Analysis

Anti-SARS-CoV-2 IgG antibodies were assessed using the commercially available Abbott ARCHITECT^®^ i2000 sr platform (Abbott Laboratories, Chicago, IL, USA). The assay is a chemiluminescent microparticle immunoassay (CMIA) for detection of IgG targeting SARS-CoV-2 nucleoprotein. According to the recommendations from ECDC, the results of Abbott ARCHITECT were confirmed by an in-house, cytopathic-effect-based virus neutralization assay (NT), using a protocol described previously [9].

### 2.5. Statistical Analysis

Baseline characteristics of participants are summarized as frequencies and proportions for categorical data and as means and standard deviations or medians and interquartile for metric data. Comparison between cohorts concerning these characteristics were dome by Fisher’s exact probability tests for categorical data and Student’s t tests or Mann–Whitney tests for metric data. Logistic regression has been performed with working environment and profession as predictors adjusted for age and gender. Missing values were excluded from analysis.

## 3. Results

### 3.1. Demographics and PCR Confirmed Infections

A total of 599 HCWs were included. Demographics of this cohort are displayed in Table 1. The median age was 40 years (interquartile range, 30–50) and 74% of the recruited HCWs were female. Of HCWs included, 23% were smokers. The most frequently included profession was nurse (45.4%), followed by physician (31.7%), secretaries and administrative staff (9.5%), physiotherapist (2.2%) and others (11.2%). HCWs working at the outpatient department accounted for 37.1%, at the ward for 46.4% and at the intensive care unit for 16.5%. HCWs were invited for a follow-up visit within two months. The median distance between both visits was 43 days (interquartile range, 35–75). In the follow-up visit, 553 of 599 HCWs were included (only 7.7% lost to follow-up). Due to screening, post-exposure or symptom-related SARS-CoV-2 testing by PCR, 432 HCWs (72.1%) had at least one PCR test result from an upper respiratory tract swab obtained during the observational period (1.84 tests per individual). In 5 HCWs, SARS-CoV-2 infection was diagnosed by PCR.

### 3.2. Analysis of Antibody-Positive HCWs

Overall, 6 HCWs tested antibody positive at the baseline visit (1%) and additional two HCWs tested positive in the follow-up visit (1.5%). In all these HCWs, presence of SARS-CoV-2-specific antibodies was confirmed by NT. Of note, only 7 out of 8 HCW tested positive by CMIA. Since the respective HCW was a PCR-confirmed case, the serum sample was additionally tested by NT. In the whole cohort, we identified a single false-positive result by first-line antibody testing (Abbott ARCHITECT^®^ positive, NT negative). Figure 1 highlights positively tested HCWs depending on their working environment and gives a hint on their history of unprotected COVID-19 contact.

Professions diagnosed with COVID-19 were mostly physicians (5 of 190, 2.6%), followed by nurses (2 of 267, 0.7%) and others (1 of 141, 0.7%) (*p* = 0.2). Half of the positively tested HCWs reported unprotected contact with COVID-19. Two positive HCWs reported unprotected contact in leisure time, one positive HCW reported a stay in a high-risk area and one positive HCW had regular contact to a positively tested family member. Furthermore, two positive HCWs reported unprotected contact during work. One positive HCW was in regular contact with a COVID-19 patient, but did not report unprotected contact. In one HCW, the source of infection was not identifiable. Only one positively tested HCW reported asymptomatic infection. Interestingly, no antibodies were detected in this HCW by CMIA, but by NT. Positively tested HCW are summarized in Table 2. Physicians were significantly more often tested than nurses (*p* < 0.001). Of note, out of the 8 HCWs with serologically confirmed SARS-CoV-2 infection, only 5 were identified by screening PCR testing.

### 3.3. Analysis of Symptoms

A total of 165 (27.5%) of the HCWs reported symptoms that could be indicative of COVID-19. The most frequently reported symptoms were headache (12.4%), congestion or runny nose (12.2%), fatigue (12.2%) and cough (11.9%). Altered taste or smell (2.2%) and fever (7.5%) were reported less frequently, but distinctly more often in positive HCWs. Furthermore, we evaluated the odds for an antibody-positive test result depending on symptoms. Altered taste or smell (OR 53, 95% CI (9.61–292.41), *p* =< 0.001), congestion or runny nose (OR 14.5, 95% CI (2.61–80.59), *p* = 0.002), fever (OR 6.57, 95% CI (1.17–36.92), *p* = 0.03), cough (OR 7.37, 95% CI (1.46–37.2), *p* = 0.016) and fatigue (OR 7.14, 95% CI (1.41–36.03), *p* = 0.017) were associated with an increased odds for COVID-19 (Appendix A).

### 3.4. Analysis of Risk Factors

In general, HCWs working at COVID-19 (146 of 205, 71.2%) wards had significantly more contact than HCWs working on non-COVID-19 units (53 of 351, 15%)(*p* < 0.001) (Appendix A). However, the risk for COVID-19 transmission did not differ between both groups (positively tested HCWs, 4 of 8 (50%); negatively tested HCWs, 201 of 591 (34%); *p* = 0.455). Unprotected contact was reported significantly more often in positively tested HCWs (4 of 8, 50%) compared to negatively tested HCWs (47 of 591, 8%) (*p* = 0.003), especially when it happened in leisure (positively tested HCWs, 2 of 8 (25%); negatively tested HCWs, 5 of 591 (0.8%); *p* = 0.003). Further, positively tested HCWs reported significantly more often a positive family member in the household (positively tested HCWs, 1 of 8 (12.5%); negatively tested HCWs, 2 of 591 (0.3%); *p* = 0.04). For travel history, we did not find a significant difference (positively tested HCWs, 2 of 8 (25%); negatively tested HCWs, 37 of 591 (6.3%); *p* = 0.09). Problems handling PPE were only reported in negatively tested HCWs (positively tested HCWs, 0; negatively tested HCWs, 30 of 591 (5%)) (Appendix A).

### 3.5. Evaluation of Safety Measures

To assess safety measures, we evaluated reasons for quarantine, transmission between HCWs and quality of communication. In total, 67 of our participants had to stay in quarantine, the main reason being contact with a COVID-19 patient (*n* = 39, 58.2%: three of them within a high-risk area), followed by a stay in a high-risk area (*n* = 15, 22.4%), COVID-19 characteristic symptoms (*n* = 6, 9%), COVID-19 (*n* = 5, 7.5%) and high-risk contact (*n* = 2, 3%). A closer look at the seropositive HCWs who did not report a history of quarantine revealed that despite all of them being tested previously by PCR, none of them was tested by PCR positively. Nevertheless, we did not report increased numbers of COVID-19 cases in their working group. Assessment of the quality of communication and COVID-19 specific training performed at our center showed that HCWs generally felt well informed and trained (highlighted in Appendix A). However, 27 HCWs reported problems, namely, lack of training (*n* = 12), defective material (*n* = 9), no suitable PPE (*n* = 3) and uncomfortable PPE usage (*n* = 3). This experience neither differed between both visits nor between professions.

## 4. Discussion

This prospective, longitudinal single-center study demonstrates that prevention strategies introduced at the General Hospital of Vienna in early March allowed safe handling of SARS-CoV-2 patients. Seroprevalence of HCWs remained stable throughout the follow-up visit in the low-risk cohort but increased in the high-risk cohort. Main risk factors for infection were unprotected SARS-CoV-2 exposure and SARS-CoV-2 infection in the family. Yet prevalence did not differ when compared to seroprevalence in the Viennese population (data not shown).

### 4.1. COVID-19 Setting and Seroprevalence Analysis

Previously published reports have estimated a 3.4-fold higher risk for frontline HCWs reporting a positive test result than the general community in the UK and USA [1]. In accordance, China and Italy reported a high proportion of HCWs among COVID-19 infections at an early stage of the pandemic [2,5]. A recently published report from the Centers of Disease Control and Prevention reported a seroprevalence of 6% in frontline HCWs in a multistate hospital network in the USA [10]. Therefore, implementation and adherence to infection control strategies are of utmost importance [10,11,12]. The COVID-19 situation reported in our hospital is exceptional due to two factors. First, we describe the situation in a COVID-19 low-incidence tertiary care center. COVID-19 patients in Vienna are distributed to hospitals according to a multi-staged plan, in which the Vienna General Hospital is primarily responsible for maintenance of medical care of non-COVID-19 patients, some of them at higher risk for severe COVID-19. Therefore, COVID-19 patients were only partially admitted under controlled conditions. Second, early implementation COVID-19 infection control measures by the Austrian government achieved a significant reduction in the daily infection rate resulting in a controlled management of COVID-19 patients. Although 205 HCWs in our study worked at COVID-19 wards, only three tested positive with suspected infection at work, whereas the remaining five patients got infected elsewhere. The seroprevalence in our cohort was 1.5%. It did not differ significantly between different risk groups. In the follow-up visit seroprevalence increased but did not exceed Viennese citizens’ general seroprevalence during the same period (data not shown). All positively tested HCWs reported symptoms. Altered taste or smell, congestion or runny nose, cough and fatigue were most often reported and were associated with the highest odds for COVID-19.

### 4.2. Handling of PPE and Risk for COVID-19 Infection

Regular training and acceptance of the methods are key factors for an efficient implementation of protocols. To evaluate potential pitfalls, we further assessed if implementation methods and the amount of training was felt to be sufficient and asked for problems noticed. HCWs generally felt well informed and trained with the use of their PPE. Most frequently mentioned criticism regarded the quality or inadequate size of provided PPE, two factors known to be essential for the prevention of respiratory illnesses [1,13]. Notably, risk factors for COVID-19 were unprotected contact, positive household contact and travel history. Unprotected contact was defined as a COVID-19 contact without proper PPE lasting more than 15 min or more within a range of two meters. However, these recommendations are based on limited data [14]. Medical masks may prevent large respiratory droplets and splashes and prevent the spreading of respiratory droplets by the person wearing them [2]. A recently published study evaluating aerosol and surface distribution emphasized a maximum transmission rate of four meters [14,15]. However, this estimate depends on specific conditions and cannot be generalized to all situations that occur in the health care setting. In our cohort, 44 HCWs reported contact with COVID-19 patients during work. Contact with COVID-19 patients was not significantly associated with a higher risk for COVID-19 compared to HCWs without unprotected contact. Three of the seropositive HCWs were not previously tested positive by PCR and continued working, and hence posed a risk for other HCWs and patients. Nevertheless, we did not discover an increased rate of infections in the follow up visit.

### 4.3. Potential Pitfalls of SARS-CoV-2 Testing

Of the HCWs, 72.1% were tested by PCR, with a mean of 1.84 tests per individual. Nonetheless, only five HCWs of our eight positively tested HCWs by CMIA were previously tested positive by PCR, suggesting difficulties in PCR testing strategies respectively sensitivity. Recent reports emphasized that pauci- or asymptomatic individuals contribute to COVID-19 transmission [16]. Furthermore, PCR is limited in describing disease burden of COVID-19; mostly, due to a low level of viral shedding during incubation and early infection, variability of sites where the virus is detectable and adequate collection of samples [17,18,19]. Hence, antibody analysis is of utmost importance. Commercial immunoassays vary in their performance, resulting in lower sensitivities in the real-life setting than the ones reported in the manufacturers’ specifications. Indeed, Abbott ARCHITECT assays failed to detect antibodies in a HCW with a PCR-confirmed SARS-CoV-2 infection, whose neutralizing antibodies were clearly detected by our NT. In an earlier evaluation at our center, Abbott ARCHITECT demonstrated a positive predictive value at lower prevalence population (1%) of only 52.3% and a lower sensitivity than reported by the manufacturer, highlighting pitfalls of antibody diagnostics [20,21].

### 4.4. Strengths and Limitations

An interesting aspect of this study is the fact that initial seroprevalence data were obtained at an early phase of the epidemic in Vienna when only four participants overall and one participant in the high-risk group tested positive. This allowed a precise calculation and description of risk-factors for the development of COVID-19. Second, we assessed various personal risk factors to minimize the confounding of our results regarding the application of PPE and our control measures. Finally, our cohort describes the use of internationally established infection control measures in a low prevalence cohort, thus reducing the influence of specific confounders such as limited hospital capacity, which lead to an imbalance in workload, distress failure and an increased risk of COVID-19 transmission.

We acknowledge several limitations. First, the questionnaire assessed risk factors, previously performed PCR tests and information about quarantine retrospectively on a self-report basis. Subsequent recalling bias might have influenced our results. Second, our cohort was not sampled randomly. HCWs were messaged by email or contacted in person. Although not intended, we acknowledge that HCWs with recent contact or suspicion of COVID-19 were more likely to participate in our study as they were more interested in the antibody results. However, this limitation is inherent as participants had to give consent voluntarily. Finally, classification into risk cohorts was based solely on environmental factors, independent of the actual frequency of SARS-CoV-2 contact among participating HCWs. This limitation was caused by early formation of cohorts at the time of our baseline visit, when most HCWs reported no contact with a COVID-19 patient at all. To account for this limitation, we assessed the frequency of SARS-CoV-2 contact inside and outside the hospital, which was significantly higher in our high-risk cohort.

## 5. Conclusions

In conclusion, the present study highlights that internationally used COVID-19 infection control measures proved feasible in reducing the risk of transmission of SARS-CoV-2 from patients and among HCWs in a low incidence hospital. However, the rate of positively tested HCWs will most probably increase within the next year. Further evaluations of our cohort are warranted.

## Figures and Tables

**Figure 1 ijerph-18-04201-f001:**
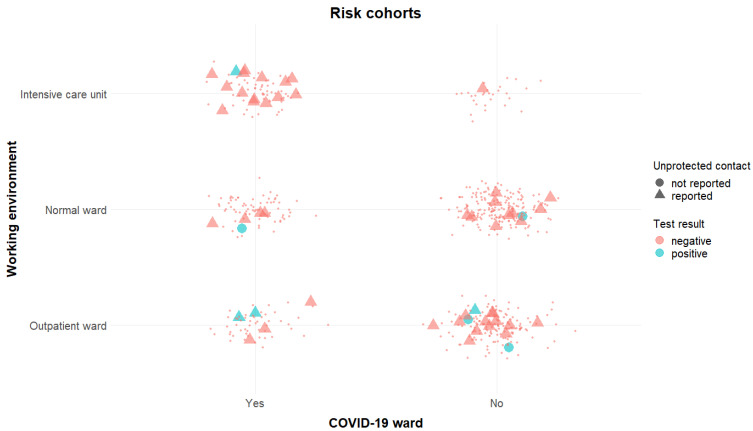
Highlights all HCW recruited in our study. Triangles highlight unprotected contact, the colour blue highlights a positive test result and a blue triangle demonstrates a positive tested HCW with history for unprotected contact. COVID-19, Coronavirus Disease 19.

**Table 1 ijerph-18-04201-t001:** The demographic characteristic of our cohort. SD, standard deviation; COVID-19, Coronavirus Disease 19; HCWs, Health Care Workers; SARS-CoV-2, severe acute respiratory syndrome coronavirus; PCR, polymerase chain reaction; CMIA, chemiluminescent microparticle immunoassay; ^a^ one HCW tested positive by PCR did not appear positive by CMIA but in neutralization test.

Demographics	High-Risk Cohort (*n* = 205)	Low-Risk Cohort (*n* = 394)	Overall (*n* = 599)
Age (mean(SD))	36.4 (10.2)	42.3 (11.7)	40.33 (11.5)
Sex (*n*(%))			
Female	139 (67.8)	304 (77.2)	443 (74)
Male	66 (32.2)	90 (22.8)	156 (26)
Smoking–yes (*n*, [%])	47 (22.9)	90 (22.8)	137 (23)
Profession (*n*, [%])			
Physicians	57 (27.8)	133 (33.8)	190 (31.7)
Nurses	114 (55.6)	158 (40.1)	272 (45.4)
Administrative staff	17 (8.3)	40 (10.2)	57 (9.5)
Physiotherapists	3 (1.5)	10 (2.5)	13 (2.2)
Others	14 (6.8)	53 (13.5)	67 (11.2)
Work environment (*n*(%))			
Outpatient department	56 (27.3)	166 (42.1)	222 (37.1)
Normal ward	79 (38.5)	199 (50.5)	278 (46.4)
Intensive care unit	70 (34.1)	29 (7.4)	99 (16.5)
Contact to COVID-19 patients (*n*(%))			
Protected contact	146 (71.2)	59 (15)	205 (34.2)
Unprotected contact	26 (12.7)	25 (6.3)	51 (8.5)
No contact reported	59 (28.8)	335 (85)	394 (65.8)
SARS-CoV-2 Tests (*n*(%))			
HCW tested	141 (68.8)	291 (73.9)	432 (72.1)
Tests per individual	1.92 (2.27)	1.8 (2)	1.84 (2.1)
Positive test results (*n*(%))			
Baseline visit (*n* = 599)			
Neutralisation test	3 (1.5)	3 (0.8)	6 (1)
PCR	3 (1.5)	1 (0.3)	4 (0.7)
CMIA	2 (1) ^a^	3 (0.8)	5 (0.8)
Follow-up visit (*n* = 553)			
Neutralization test	4 (2)	4 (1)	8 (1.5)
PCR	0	1 (0.3)	1 (0.2)
CMIA	3 (1.5)	4 (1)	7 (1.3)

**Table 2 ijerph-18-04201-t002:** All positive tested HCW at our center by neutralization test. m, male; f, female; COVID-19, Coronavirus Disease 19; HCWs, Health Care Workers; SARS-CoV-2, severe acute respiratory syndrome coronavirus; PCR, polymerase chain reaction; CMIA, chemiluminescent microparticle immunoassay.

Positive Tested HCW	HCW 1	HCW 2	HCW 3	HCW 4	HCW 5	HCW 6	HCW 7	HCW 8
Demographics	m, 35a	f, 59a	f, 42a	m, 31a	m, 29a	m, 41a	f, 60a	m, 45a
Profession	nurse	secretary	nurse	physician	physician	physician	physician	physician
COVID-19 contact	Work (daily)	Leisure (daily)	leisure (once)	none	work	none	leisure and work	work
Unprotected contact	none	none	yes (leisure)	none	yes (work)	none	yes (leisure)	yes (work)
Suspected source of infection	work	leisure	leisure	unknown	work	leisure	leisure	work
Number of PCR tests performed	one	two	two	one	three	one	four	three
SARS-CoV-2 PCR results	negative	positive	positive	negative	positive	negative	positive	positive
SARS-CoV-2 CMIA results	positive	positive	positive	positive	positive	positive	positive	negative
Reported symptoms	sniff, headache, body aches	fever, cough, sniff, fatigue, shortness of breath	fever, cough, sore throat, fatigue, smelling problems	sniff, smelling problems	headache, cough, sniff, fatigue, smelling problems	fever, headache, cough, body aches, fatigue, smelling problems	fever, headache, cough, fatigue, smelling problems	asymptomatic

## Data Availability

The data presented in this study are available in the Appendix A.

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
