# Peer review of "A Longitudinal Seroprevalence Study Evaluating Infection Control and Prevention Strategies at a Large Tertiary Care Center with Low COVID-19 Incidence"

_ijerph, 2021, doi:10.3390/ijerph18084201_

Round 1
Reviewer 1 Report
There are a couple of issues that deserve a further attention by the authors, and among them I would like to underline the following:
- The title looks quite lenghty
- There is a lack of a theoretical reasoning and a clear working hypothesis in the background.
- No any reference to the people who were eligible to participate in the study and there is a lack of the study's flow chart.
- The definion of high-risk group is based only on envronmental factors. A discussion on its potential expansion could be importnat and it could be placed in the discussion section of the paper.
- Some meaninguful subheadings could be added in the discussion section.
- The STROBE Statement could be completed and submitted with the revised manuscript.
Author Response
Dear reviewer,
our team read with much interest your recommendations regarding our manuscript.
- The title looks quite lengthy
Thank you very much for your comment. We shortened the title of the manuscript to “Evaluation of infection control and prevention strategies at a large tertiary care center with low COVID-19 incidence“. We believe it is now more precise.
- There is a lack of a theoretical reasoning and a clear working hypothesis in the background.
- No any reference to the people who were eligible to participate in the study and there is a lack of the study's flow chart.
Thank you very much for your comment. Our analysis was performed in HCWs at the General Hospital Vienna. At the start of the study HCWs were informed by email about the possibility to participate at our study. Beside age (≥18 a) we had no restriction criteria to participate at our study. Further, 553 of 599 HCWs participated at the second visit as they did not show up and follow-up approach was not successful. We are now describing selection of participants more precisely. Concerning the mentioned flow chart. We primarily intended to implement a flow chart, but worried it would not provide much additional information. However, if you believe it would improve the comprehension of the text we are pleased to do so.
- The definion of high-risk group is based only on envronmental factors. A discussion on its potential expansion could be importnat and it could be placed in the discussion section of the paper.
- Some meaninguful subheadings could be added in the discussion section.
Thank you very much for your suggestion. We added subheadings into our discussion section.
- The STROBE Statement could be completed and submitted with the revised manuscript.

Reviewer 2 Report
A well done article whose topic its of major interest not only in this period. Well done and contratulations!
Author Response
Dear reviewer,
thank you very much for your kind words.
Best regards,
Lorenz Schubert and Selma Tobudic
Reviewer 3 Report
Review for IJERPH-1160491: A longitudinal Seroprevalence Study in Health Care Workers 2 of a Large Tertiary Care Center With Low COVID-19 Incidence 3 Reveals High Efficacy of Prevention Strategies
I appreciate the invitation to review this manuscript that addresses an important public health issue. The manuscript is original and has been well-designed methodologically. However, the manuscript presents a some issues that have to fix or address before publication.
- Introduction
Line 44: please specify the IPC meaning.
Line 47: virus transmission instead of disease transmission.
Line 53: in Supplementary Table S1, please specify how long each control measure lasted in Austria. Did this duration cover the entire study period of this study?
Lines 49 to 55: please provide a reference for this paragraph.
- Materials and Methods
2.1. Study Design and Participants
Line 73: the term efficacy does not fit what was done in the study. I strongly recommend that the authors avoid using the term efficacy in this manuscript (throughout the entire text), since the evaluation of effectiveness presupposes the existence of a control group in order to assess an intervention. As described in item 2.3. Formation of Cohorts and Assessment of Risk Profile, the study set out to assess profiles and risk factors for SARS-CoV-2 infection among health care workers. The authors must also be aware of possible confounding factors, since it is impossible to control all the other factors that can influence the occurrence of new infections by SARS-CoV-2. Therefore, it is not feasible to evaluate the isolated effectiveness of the correct use of PPE.
Considering all these insights, I also suggest that the title of the manuscript be revised.
2.5. Statistical Analysis
Line 113: were done instead were dome.
Lines 114 – 115: what were the strategies for assessing the presence of confounding factors? How the model fitness was assessed?
Please inform which software was used for the statistical analysis, including the one used for preparing the figures.
- Results
3.1. Demographics and PCR Confirmed Infections
Line 120: please avoid starting sentences with numbers or percentages.
Table 1 should provide the p-values for the statistical hypothesis testing described in lines 112 – 114.
Were there any follow-up losses in the study? This is not clear from the results.
- Discussion
Lines 199 – 202: authors should consider and emphasize that the burden and incidence of COVID-19 in the study area is low (including the whole population of Austria), therefore, it is expected that few new cases will occur in this period, since the population has adhered to the implemented COVID-19 control strategies by the Austrian government.
Lines 270 – 272 (“Second, we assessed various personal risk factors to minimize the confounding of our results regarding the efficacy of protection based on usage of PPE and our control measures”): the authors should describe it in Results and in the Methods because there is no results neither for the assessment of confounding and for model fitness.
Author Response
Dear reviewer,
our team read with much interest your recommendations regarding our manuscript.
- Introduction:
1.1. Line 44: please specify the IPC meaning.
Thank you very much for your comment, we spelled out IPC.
1.2. Line 47: virus transmission instead of disease transmission.
Indeed, virus transmission is much more precise. Thank you very much!
Line 53: in Supplementary Table S1, please specify how long each control measure lasted in Austria. Did this duration cover the entire study period of this study?
Thank you very much for your comment. We specified the duration of implemented control measures. Indeed, hospital specific measures covered the whole study period. However, national wide strategies changed throughout the study period.
Lines 49 to 55: please provide a reference for this paragraph.
Thank you very much for your comment, we provide now a reference for the paragraph.
- Materials and Methods
2.1. Line 73: the term efficacy does not fit what was done in the study. I strongly recommend that the authors avoid using the term efficacy in this manuscript (throughout the entire text), since the evaluation of effectiveness presupposes the existence of a control group in order to assess an intervention. As described in item 2.3. Formation of Cohorts and Assessment of Risk Profile, the study set out to assess profiles and risk factors for SARS-CoV-2 infection among health care workers. The authors must also be aware of possible confounding factors, since it is impossible to control all the other factors that can influence the occurrence of new infections by SARS-CoV-2. Therefore, it is not feasible to evaluate the isolated effectiveness of the correct use of PPE.
Thank you very much for your comment. As noted from you precisely, a control group is obligatory for efficacy evaluations of interventions. Hence, we adapted the text by avoiding the terms effectiveness and efficacy. Further, we tried to cover a lot of confounding factors. However, we are aware that it is not possible to include all of the factors in the analysis.
2.2. Considering all these insights, I also suggest that the title of the manuscript be revised.
Thank you very much, we adapted to title of the manuscript to “Evaluation of infection control and prevention strategies, a longitudinal seroprevalence study at a large tertiary care center with low COVID-19 incidence.”
Thank you very much for your considerate suggestions.
We looking forward hearing from you.
Lorenz Schubert and Selma Tobudic
Reviewer 4 Report
Thank you for the opportunity to review this manuscript. It describes the evaluation of infection control measures to prevent nosocomial SARS-CoV-2 transmission at the Vienna General Hospital. Although, the study is well conducted, its added knowledge is a bit limited. The authors concluded that the prevention strategies proved highly effective, because the overall seroprevalence of the Vienna General Hospital does not exceed the seroprevalence of the general population. However, one has to keep in mind that the study contains of only one data point (i.e., the Vienna General Hospital) as evidence. Thus, the conclusion is a bit far-fetched and needs to be downturned. The manuscript contains some other findings that are interesting but not completely new. For instance, the symptoms associated with COVID-19 (e.g., Lan et al., 2020; La Torre et al., 2020; Magnavita et al., 2020) or the association between family member infections and HCW infections (Craxford et al., 2021; Lorenzo & Carrisi, 2020) have been shown by others. Nevertheless, as the study is well conducted, there is nothing fundamental to hold up a publication over (after the conclusion is downturned).
Minor points
- Please always report exact p-values unless p < .001.
- Please report the frequencies and percentages in section 3.3. Analysis of Risk Factors.
- Reporting of percentage when discussing the proportion of 8 cases is not necessary (section 3.2. Analysis of Antibody-Positive HCWs).
- Please delete (p. 6 row 195-197) “This section may be divided by subheadings. It should provide a concise and 195 precise description of the experimental results, their interpretation, as well as the experi-196 mental conclusions that can be drawn.”
References:
Craxford, S., Nightingale, J., Ikram, A., Marson, B. A., Kelly, A., Norrish, A., ... & Ollivere, B. (2021). SARS-CoV-2 transmission from the healthcare setting into the home: a prospective longitudinal cohort study. medRxiv. https://www.medrxiv.org/content/medrxiv/early/2021/02/03/2021.02.01.21250950.full.pdf
Lan, F. Y., Filler, R., Mathew, S., Buley, J., Iliaki, E., Bruno-Murtha, L. A., ... & Kales, S. N. (2020). COVID-19 symptoms predictive of healthcare workers’ SARS-CoV-2 PCR results. PloS one, 15(6), e0235460.
La Torre, G., Massetti, A. P., Antonelli, G., Fimiani, C., Fantini, M., Marte, M., ... & Mastroianni, C. M. (2020). Anosmia and ageusia as predictive signs of COVID-19 in healthcare workers in Italy: a prospective case-control study. Journal of Clinical Medicine, 9(9), 2870.
Lorenzo, D., & Carrisi, C. (2020). COVID-19 exposure risk for family members of healthcare workers: An observational study. International Journal of Infectious Diseases, 98, 287-289.
Magnavita, N., Tripepi, G., & Di Prinzio, R. R. (2020). Symptoms in health care workers during the COVID-19 epidemic. A cross-sectional survey. International journal of environmental research and public health, 17(14), 5218.
Author Response
Dear reviewer,
Our team read with much interest your recommendations regarding our manuscript.
- The authors concluded that the prevention strategies proved highly effective, because the overall seroprevalence of the Vienna General Hospital does not exceed the seroprevalence of the general population. However, one has to keep in mind that the study contains of only one data point (i.e., the Vienna General Hospital) as evidence. Thus, the conclusion is a bit far-fetched and needs to be downturned.
Thank you very much for your comment. We downturned the conclusion of our manuscript, and avoided the words efficacy and effectiveness throughout our manuscript.
- Please always report exact p-values unless p < .001.
Thank you very much, we adapted the reporting of p-values throughout the manuscript.
- Please report the frequencies and percentages in section 3.3. Analysis of Risk Factors.
Thank you very much, we adapted the paragraph 3.3 by naming frequencies and percentages of reported risk factors. We did not show them primarily as they were all listed in supplementary table 7 and we wanted to facilitate reading of the manuscript.
- Reporting of percentage when discussing the proportion of 8 cases is not necessary (section 3.2. Analysis of Antibody-Positive HCWs).
Thank you very much for the comment, we deleted the percentages in the section 3.2.
- Please delete (p. 6 row 195-197) “This section may be divided by subheadings. It should provide a concise and 195 precise description of the experimental results, their interpretation, as well as the experi-196 mental conclusions that can be drawn.”
Thank you very much for the recommendation. We adapted the first paragraph of the discussion section, focusing on experimental results and a short interpretation.
Thank you very much for your considerate suggestions.
We looking forward hearing from you.
Lorenz Schubert and Selma Tobudic
Round 2
Reviewer 1 Report
The revised manuscript has much improved under the light of the last editorial's commentary.